# The Dilemma of Compulsory Vaccinations—Ethical and Legal Considerations

**DOI:** 10.3390/healthcare11081140

**Published:** 2023-04-15

**Authors:** Yael Sela, Keren Grinberg, Rachel Nissanholtz-Gannot

**Affiliations:** 1Department of Nursing Sciences, Ruppin Academic Center, Emek- Hefer 4025000, Israel; kereng@ruppin.ac.il; 2Department of Health Systems Management, Ariel University, Ariel 40700, Israel; rachelng@ariel.ac.il; 3Myers-JDC-Brookdale Institute, Jerusalem 9103702, Israel

**Keywords:** compulsory vaccination, public health, law, immunization

## Abstract

The high childhood vaccination coverage in Israel leads to a low rate of morbidity from the diseases against which the vaccination in administered. However, during the COVID-19 pandemic, children’s immunization rates declined dramatically due to closures of schools and childcare services, lockdowns, and guidelines for physical distancing. In addition, parents’ hesitancy, refusals, and delays in adhering to routine childhood immunizations seem to have increased during the pandemic. A decline in routine pediatric vaccine administration might indicate that the entire population faces increased risks for outbreaks of vaccine-preventable diseases. Throughout history, vaccines have raised questions about their safety, efficacy, and need among adults and parents who feared or hesitated to vaccinate their children. Objections derive from various ideological and religious reasons or concerns about the possible inherent dangers. Mistrust in the government and/or economic or political interests also raise concerns among parents. The importance of providing vaccines to maintain public health, as opposed to the autonomy of the individuals over their body and their children, raises ethical questions. In Israel, there is no legal obligation to get vaccinated. It is imperative to find a decisive solution to this situation without delay. Furthermore, where democratically one’s principles are sacred and where one’s autonomy over one’s body is also unquestionable, such a legal solution would not only be unacceptable but also rather impossible to enforce. It seems that some reasonable balance between the necessity to preserve public health and our democratic principles should apply.

## 1. Introduction

The eradication of epidemics, or at least reducing their rate, depends on a wide range of factors, the main of which is strengthening the infrastructure of public health systems and public medicine, as well as environmental and social variables that affect health [1]. It has been proven that vaccinations as such are efficient means to minimize the spread of epidemics and play an important role in the efforts to eliminate them completely [1,2].

Since the outbreak of the COVID-19 pandemic, global childhood vaccinations have experienced the largest sustained decline in about 30 years [2,3,4]. 

According to new data from the Federal Center for Disease Control and Prevention, the percentage of US children entering kindergarten with their required immunizations fell to 93% in the 2021–2022 school year, 2 percentage points below the recommended herd immunity level of 95% and lower than vaccination rates in 2020–2021. After the pandemic, those historically low rates worsened [5]. As of the end of 2020, the State of Israel, with a population of 9.3 million, had administered more COVID-19 vaccine doses than all countries aside from China, the US, and the UK [6]. In contrast to the success in vaccinating the population against COVID-19, there was a decrease in all routine childhood vaccinations compared to the period before the epidemic [7]. 

Consequently, we face increasing dangers of children contracting diseases, which had been considered eliminated or under control in big parts of the world, provided vaccines had been administrated regularly. Diseases, such as measles, could easily spread and become a worldwide epidemic [8]. Therefore, this article deals with tactics and strategies necessary to build up the defenses against such possibilities. 

## 2. Vaccination Rates in Israel

Routine immunization coverage has been high in the State of Israel over the years. However, current data show that in the last two years, there has been a certain decrease in immunization rates, between 0.5% and about 2%, mainly in the booster doses [7], and especially in all routine vaccinations for babies, compared to the period before the COVID-19 pandemic. These include vaccines for diseases, such as measles, mumps, and rubella, that are given routinely at the age of one month, two months, four months, and six months and vaccines for chickenpox, whooping cough, and *Haemophilus influenzae*. However, it should be noted that the Ministry of Health in Israel does not have central statistics and usually presents data on vaccination rates in districts or regions.

In the Jerusalem district, there is a more substantial decrease than in other districts. For example, between the years 2017–2021, the district recorded a decrease of about 7% in the administration of the pentavalent vaccine, which is given against dysentery, tetanus, and whooping cough. In the Tel Aviv district, a decrease of about 5% was also recorded during the same period. In addition, in vaccinations given against measles, mumps, and rubella, there was a decrease of about 3% from 99% in 2017 to 96% in 2021. An additional 5% decrease in the administration of polio vaccines from 97% in 2017 to 92% in 2021 was recorded [7]. There was also a decrease in vaccination against the *Haemophilus influenzae* bacterium, which causes severe meningitis in infants. A 7% decrease was recorded in the Jerusalem district and a 5% decrease in the Tel Aviv district. The Ministry of Health warned that although most of the technical limitations of the epidemic have been removed, the trend of decreasing immunization coverage among children continues. This decrease may have epidemiological implications, such as damage to herd vaccination and an increased risk of resurgence of certain diseases [7].

### 2.1. Vaccination Hesitancy

The drop in childhood immunizations is attributable, in part, to the pandemic that has caused supply-chain disruption, diversion of resources, and lockdowns, which have limited immunization services. Economic challenges have also decreased some government financing of routine vaccinations [9,10,11].

Public experts claim that the politicization of public health and increasing distrust of government have skewed parents’ previously positive attitudes about vaccines for measles, mumps, and rubella [12]. In Israel, the discussion of general vaccinations of children against diseases, such as measles, mumps, and chickenpox, has emerged. An evaluation of changes in parents’ approach to vaccination during COVID-19 found that regarding the relationship between stress levels and the change in parents’ attitudes about vaccination after the pandemic, parents who had vaccine hesitations before were undecided now [13]. Parents’ hesitancy, refusals, and delays in adhering to routine childhood immunizations seem to have increased during the pandemic [8]. The strengthening of antivaccination movements in recent decades has coincided with unprecedented increases in the incidence of some communicable diseases [14].

This delay or refusal of vaccination is termed vaccine hesitancy (VH). VH is of grave concern, such that it was listed by the WHO as one of the ten threats to global health in 2019 [15]. The concept of vaccine hesitancy, defined by the World Health Organization as “delay in acceptance or refusal of vaccines despite availability of vaccine services” is used to describe the continuum of concerns regarding vaccine acceptance—ranging from worries about vaccine safety to antivaccination sentiment.

A survey of parents’ attitudes conducted by the Israeli Ministry of Health shows that there has been a dramatic and clear increase in the rate of those exposed to antivaccination claims in the media and social networks in recent years—from 59% in 2016 to 77% in 2022 [7]. In addition, 30% of parents express only partial trust in the Ministry of Health, and about a tenth of parents see vaccines as potentially harmful to children. The increased hesitation causes serious concern in the Ministry of Health regarding an immediate outbreak of diseases and raises questions in relation to the right ways to increase immunization coverage [7].

It is often the case that when parents are grouped together in a certain area, and their children catch measles, they infect others. This process creates pockets of unvaccinated populations. A “pocket” is a defined and limited geographical area, for example part of a neighborhood, a school, or a kindergarten, where the immunization coverage rate is lower than 95%. The risk of pockets of non-vaccination is in decreasing the rate of immunization, significantly endangering herd immunity, potentially spreading the disease, and harming public health [16]. For example, the herd immunity level, i.e., the rate of immunity coverage that minimizes the chances of a disease outbreak, even among those who were not vaccinated for measles, which is highly contagious, requires 95% of the population to be vaccinated. The existence of pockets of nonimmunization proved fateful in many recent outbreaks in closed communities [17,18,19].

Major groups that are recognized in Israel as hesitant and objecting to vaccines are the middle- to high-economic status groups, which reject vaccinations on ideological or philosophical grounds. In 2007, the Tel Aviv-Yafo Municipality conducted a survey to assess the scope of the phenomenon. The findings indicated that 5% of the vaccination potential of the 2004 cohort had not been realized. By socioeconomic analysis, among children from middle- to high-economic status groups, about 3% of vaccinations were not performed because of the parents’ ideological or philosophical views, whereas among children from low socioeconomic groups, only about 0.5% stated these reasons. The main grounds for nonvaccination or partial vaccination among this group were forgetfulness, unawareness, or failure to understand the importance of vaccination [20].

In a survey conducted in Israel, most parents (90%) declared that their children received all the routine vaccinations, similar to the situation in the US. However, the same survey found that only 32% of the parents reported that they did not fear vaccinations at all. Like in the US, parents in Israel quoted a variety of concerns about vaccinations. For instance, many parents (40%) feared “permanent damage” to their children, while 10% of the parents were concerned about “damage to the children’s immune system” [21]. Some parents believe that they can protect their children from exposure to a contagious disease through hygiene, good nutrition, and a healthy lifestyle and have a preference for natural healing methods and believe that natural immunization (allowing a child to contract the disease) is preferable to getting an artificial vaccine. It is obvious that even parents who do vaccinate their children regularly have doubts about the effectiveness of vaccines. Misinformation that is often contradictory and not necessarily accurate is being disseminated on social media platforms, which may significantly impact an individual’s decision to get vaccinated [7,12].

A less frequently discussed reason for hesitancy is country of vaccine origin. The availability of foreign-made vaccines can reinforce the belief that vaccines are not trustworthy and are part of a larger conspiracy. For example, in the USA, US-made vaccines were more trusted than those made in China or foreign-developed ones. Most Chinese (64%) expressed no preference for domestic- or foreign-made vaccines [22,23].

### 2.2. The Dilemma—Maintaining Public Health vs. Individual Autonomy

The ethical questions surrounding vaccinations have always existed. The COVID-19 pandemic has, on one hand, highlighted the importance of vaccines as part of an attempt to eradicate the epidemic. Previous studies reported that infectious outbreaks, such as COVID-19, have the potential to affect what has been achieved by immune coverage, and other studies testified that routine vaccinations and vaccines in general reduce the incidence of diseases in general. The ongoing COVID-19 pandemic is a reminder of the importance of vaccination [2,11]. However, on the other hand, it has increased opposition, which is grounded in the fear that the vaccine might be unsafe because it was developed too quickly and lacks a long-term perspective and assessment [24], which may lead to mistrust of present-day vaccines. In addition, it raises the question of the economic interests that pharma companies have with the government [25]. Some parts of the public cannot be vaccinated for various reasons, and some may have adverse reactions to the vaccine.

The balance between the desire to preserve people’s personal autonomy and their right to choose what will or will not be performed on their body and protecting at-risk populations on the other hand places complex dilemmas and challenges before health systems. The first legal solution to the vaccination problem was the United Kingdom Vaccination Act (1853), which made it compulsory for all children to be vaccinated against smallpox during their first three months of life. Parents who failed to have their children vaccinated would be subject to a fine. The Vaccination Act was a political innovation, which authorized the government to interfere in citizens’ civil rights and freedom of choice in the name of public health [26]. Antivaccination sentiments incited opposition all over Europe. Ethical questions arose with the first laws that were passed. When the Vaccination Act was extended in 1867 to include all children up to the age of 14, the Anti-Vaccination League was established, and in 1879, the London Society for the Abolition of Compulsory Vaccination started publishing the ‘Vaccination Inquirer’ [27,28]. The demonstrations and violence prompted the British Parliament to cancel the accruing fines against antivaccinationists, and it published a new act that gave parents the option not to vaccinate their offspring. The Vaccination Act was finally abolished in 1946 by the British Parliament, including compulsory smallpox vaccinations [29]. In recent years, a concerning decrease in public confidence in science has been observed, and a strong antiscientific counterculture has emerged [1]. Based on their own sources of information and selectively ignoring research that contradicts their beliefs, many laypeople believe that they have the background to challenge established scientific facts, and there is no indication that this trend will slow or reverse in the near future [30].

The inevitable questions are whether to compel parents by law to vaccinate their children and which are the best tools that the state can use to increase vaccination rates. First, we present the vaccination policy.

## 3. Vaccination Policy and Law

### Legal Solutions to Antivaccination

The state’s obligation to safeguard public health includes its authority to pass regulations that restrict its citizens and infringe on their rights for the greater good. This is especially significant in the area of vaccinations, where the behavior of individuals affects the general population’s risk. On the one hand, parents want the state to protect them and their welfare, and see the state as being responsible for public health, and, on the other hand, object to coercion and enforcement measures.

In the US, a law that compels vaccinations for the entire population was passed in Massachusetts as far back as 1807 [31], which instigated antivaccination movements that opposed the law in the name of basic civil freedoms. The struggle was conducted in the courts and on the streets through demonstrations and riots. The organized opposition managed to abolish the law in several states.

At the beginning of the 1980s, all the states in the US required a certificate of vaccinations when enrolling children in kindergarten and school. However, there is a mechanism in place that provides exemptions for health, religious, and philosophical reasons. Thirty-one states have explicit laws that a school may remove children who were exempted from vaccination in the case of an epidemic or another emergency, and in some states, vaccination can be coerced [32]. Modifying state childhood vaccination exemption laws through legislation including removing nonmedical exemptions or making them more difficult to obtain is one frequently proposed strategy for increasing vaccination rates. Major national medical organizations have endorsed this strategy for several years now [33].

In the case of the USA, an antivaccine movement that began with false assertions linking vaccines to autism accelerated roughly a decade ago in Texas around a libertarian framework known as health freedom [34]. The European Observatory on Health Systems and Policies [35] prepared a report of systematic reviews on health system-related factors influencing vaccine uptake in the European Union. Of the 28 countries, nine (Bulgaria, Croatia, Czech Republic, France, Hungary, Italy, Poland, Slovakia, and Slovenia) have legislation that compels child vaccination, and in the remaining 19, vaccination is voluntary, although in some entering the education system is conditional on a vaccination certificate.

In Canada, there is a national recommended immunization program. The laws that obligate immunization change from province to province. According to the data, only three provinces require an immunization certificate as a precondition to entering school. In Alberta, in addition to preventing the entry of unvaccinated children to school or kindergarten, parents who do not vaccinate their children on time are subject to fines and possible suspension from school [26]. In Ontario and New Brunswick, children’s acceptance to school is conditional upon parents providing proof that their children are vaccinated and is consistent with the national program (diphtheria, tetanus, polio, measles, mumps, rubella, meningococcal disease, pertussis, and varicella) [36].

In Australia, new legislation on immunization dictates steps to encourage the population to be vaccinated by means of education, financial incentives, and reduced childcare costs [37,38]. On the other hand, children’s allowances can be reduced by AUD 15,000 per family if they fail to vaccinate their children. This is aimed primarily at low-income families to encourage them to vaccinate their children. In Victoria, proof of vaccination must be presented when enrolling children in primary school, and in New South Wales, this requirement has been extended to high-school enrollment [37,38].

Similarly, in Slovenia, parents who do not vaccinate their children can expect heavy fines of up to EUR 500 per vaccination [39]. In the Czech Republic, legislation is in place that prohibits unvaccinated children from enrolling in kindergarten. Kindergarten service providers can be fined up to EUR 20,000 if they accept an unvaccinated child. Sweden has a national immunization program, and the county councils are obligated to offer the vaccinations included in the program or even add to them. However, in a report submitted to the European Commission (2018), the Swedish government stated that vaccinations were voluntary and that there were no sanctions in place for nonvaccination [40]. In Italy, there is a policy of compulsory vaccinations and required vaccinations. The program provides education and information, on the one hand, and, on the other hand, sanctions are enforced against physicians who do not comply [41], but, during the COVIC 19 pandemic, Italy’s Constitutional Court upheld the vaccine mandates introduced in 2021. The new government scrapped a COVID-19 vaccine mandate for health workers [42].

In Israel, there is no legal obligation to be vaccinated. Moreover, vaccinations, like any other medical treatment, require informed consent. However, the 1940 Public Health Ordinance (section 19) states that: “Any city, village or region in which an infectious disease has become or may become an epidemic, or that an epidemic exists near them… the director or government physician may decide to take any measures that he sees fit to protect the residents from infection. To this end, he is entitled to impose compulsory vaccination on the resident. Anyone knowingly refusing vaccination. will be charged with an offense, and is legally liable to pay a fine or be incarcerated for a period of up to one month…” [43]. As stated, the authorized party can coerce a population group to be vaccinated in cases of an infectious disease. Clearly, this relates to specific emergencies, and routine vaccinations are not included.

In 2009, an attempt was made to pass a bill that amended the 1995 National Insurance Law, which determined that children’s allowances would be reduced for families that did not routinely vaccinate their children. This bill, which tried to indirectly enforce vaccinations, was cancelled after a few years [44]. This reinforces the argument that legislation on this topic is ineffective and other ways should be utilized to raise awareness of and compliance with vaccinations. Moreover, the use of legislative means to force the public to get vaccinated could seriously damage the delicate fabric of relations between the public and the health care establishment. The Advisory Committee on Infectious Diseases and Vaccinations in the Ministry of Health discussed the public response to routine vaccinations and especially the idea to demand vaccination certificates for children before enrolling in day care facilities, but the conclusions were that this way was not effective.

Despite the conclusion that legislation and coercion were generally ineffective, on 1 January 2019, bill #833—aiming to amend the Public Health Ordinance—was placed before the Knesset [45]. The bill’s goal was to make official the “steps that are intended to increase public awareness of the need to routinely vaccinate children, and to allow the Ministry of Health to be organized to treat children and anyone in an educational institution in cases of outbreak of infectious disease”. The bill emphasized every resident’s right to vaccinate their children as outlined by the Ministry of Health, and the ministry’s obligation to provide information about these vaccinations. It also suggested establishing a database of all vaccinations performed in Israel (not just children’s vaccinations) and how they are to be managed. In addition, details were given concerning reasons to refuse vaccinations, and the Minister of Health was authorized to take extreme measures in the case of epidemic outbreaks, including coercing vaccinations and shutting down education institutions. The Knesset website reports that the bill is currently up for the second and third readings [46].

The Israeli Ministry of Health is taking several steps to improve immunization coverage among certain population groups by adjusting and making the vaccination setup accessible to their needs [7]. The steps primarily include informing parents, making house calls, and giving parents the option to prepare a personal vaccination program. In cases of infectious diseases outbreaks, the ministry relies mainly on these actions to reach populations that had not been vaccinated in the past and to try to convince them to do so. It is also worth noting that when they wanted to increase the percentage of vaccinated children, local authorities took action by distributing small incentives, such as special food and small gifts. However, because of the increase in antivaccinationists in the general population, this issue is once again on the agenda, and it is expected that discussions will be held on the balancing of values, such as duty, social responsibility, public health, and individual health, and values of autonomy, individualism, and freedom of choice [1,7,27].

There are several reasons for not getting vaccinated. It can be a medical reason or fear of vaccination because of a past experience. However, there are people who refuse vaccines because of ideological reasons. Forceful coercion of vaccinations to a population that is not willing to do so for ideological reasons raises the dilemma between respecting a person’s autonomy, i.e., their right to decide what is and is not performed on their body, and preserving public health [47]. The idea behind that thinking is that your right, as a human being, to autonomy is equivalent to your duty, as part of society, to get vaccinated to reduce the danger to yourself and your surroundings.

Compulsory vaccination—whether indirectly by cutting allowances or directly by criminal punishment or physical coercion—damages one’s right to act according to one’s principles. Physical coercion also damages one’s autonomy over one’s body.

Another argument that is often heard is the ‘hitchhiker argument’, namely the possibility that certain people can avoid vaccination based on the fact that their neighbors are vaccinated, because if not, nonvaccination would be much riskier. In fact, this is a form of taking advantage at the expense of someone else.

An important principle to be emphasized in the discussion of vaccines is justice. The popular attitude in Israel is that every individual has the basic right to a vaccine that will prevent illness. On the other hand, the government is obligated to provide vaccines for various illnesses. Indeed, the routine vaccination rate in Israel is among the highest in the world. In order to achieve substantial justice, people who cannot be vaccinated must be afforded a low-risk environment, and this is an obligation that applies to everyone, even to those who avoid vaccinations due to concerns or ideological perceptions.

The question arises, is it possible in principle to sanction those who refuse vaccination or to force them to do so? One of the main dilemmas related to the vaccination policy is whether it is moral and right to legally compel vaccinations and under which conditions this could be performed. Is the greater good or the individual child’s good superior to values, such as autonomy, individualism, and the right to physical property? This ethical question becomes even sharper when discussing it in light of relevant ethical codes, such as ADVANCE Code of Conduct for collaborative vaccine studies [48]; UNESCO universal declaration on bioethics and human rights [49]; and even the Nuremberg Code [50], which introduces the importance of maintaining human freedom and not violating the basic human rights enshrined in them. One of the core principles stated in the Nuremberg Code—informed consent—supports the argument that parents should have more freedom to choose whether to vaccinate their children. This seems to be a reference to the Nuremberg Code, which says that a subject’s voluntary consent is essential in experiments [50].

We believe that, in such cases, a balance should be found between the individual’s right to autonomy and his or her duty, as part of society, to preserve public health. Therefore, the risk of nonvaccination, the fear of infection, and its scope and severity, must be considered. A similar form of balance exists in various legal–medical issues, such as disclosure of a patient’s name by an ethics committee, thus breaching patient confidentiality, which is legally binding, if “the patient has been given the opportunity to be heard, that handing over his medical information is essential to protecting the other’s health or public health, and that the need to hand it over supersedes the issues of not handing it over” [51].

## 4. Recommendations

The dilemma between the values of patients’ individual rights and autonomy in medical procedures and preserving public health in general is one of the main questions in vaccination policy. On the one hand, everyone has the right to decide how to manage his or her health, especially when it concerns inserting a foreign substance that may or may not have side effects. It seems that the right over our bodies is an undisputed basic right. On the other hand, public health is a prerequisite of personal health, and maintaining public health requires widespread vaccination in order to achieve ‘herd immunity’—a situation in which the community is sufficiently immune, so that the outbreak of infectious disease cannot continue to spread.

## 5. Conclusions

It seems that, in Israel, achieving a balance between these values—through increased public trust—is an important step in alleviating concerns about vaccinations. Trust relates primarily to the transparency of the process, openness of the vaccine’s goals, accessibility to the information adapted to the various populations, and full information on possible side effects. Trust-promoting information requires attention to issues that concern the public, not necessarily scientific issues that experts might believe to be important. Vaccination hesitancy is often not about a lack of information but rather a surplus of information from varied (and frequently contradictory) sources and attempts to deal with conflicts that vary from vaccine to vaccine, such as vaccine safety and efficacy vs. the dangers of the disease and personal interests vs. contribution to the community [1].

## Data Availability

The data presented in this study are available on request from the corresponding author.

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
