# Peer review of "The Dilemma of Compulsory Vaccinations—Ethical and Legal Considerations"

_healthcare, 2023, doi:10.3390/healthcare11081140_

Round 1

Reviewer 1 Report

  1. The authors might elaborate more on the public health considerations in the text. Those explanations will enrich point 2.2.
  2. Two additional points may also be discussed: I noticed that the US Government has already reduced the non-medical reasons for vaccine exemptions. During the pandemic in Italy, the court decision pushed the Italian Government to withdraw the vaccine mandate (in the text, this relate to line 216-218).
  3. There is also a strong argument regarding the anti-science group's role in opposing the vaccination program, especially in the US. The author might also add some comments on this issue.
  4. Line 256-267 needs references.
  5. Was it suitable to relate The Nuremberg Code to this issue (line 294-295)? Nuremberg Code is more suitable for human research. There might be a few similar codes for medical practices outside the trials.  

Reviewer 2 Report

Line 33-35- “ It has been proven that vaccinations as such are efficient means to minimize the spread of epidemics, and play an important role in the efforts to eliminate them completely” – please include the references supporting this statement.

 Organism’s name should always be italicized eg: line 69, Haemophiles influenzae

‘In Ontario and New Brunswick, children's acceptance to school is conditional upon parents providing proof that their children had been vaccinated and is consistent with the national program’ Please include the relevant references supporting this statement.

Some data/information missing citation, please add the citation details/relevant references to corresponding data/information.

Have you considered in your essay how the imported vaccines from foreign countries’ manufacturer might affect the attitude of individuals towards vaccination, particularly in the context of anti-vaccination movement?

There are a few grammatical, typo mistakes and formatting which need to be addressed.

Some references need to be formatted.

Reviewer 3 Report

Yael Sela et al. describe the question between vaccination and freedom for a person to decide for himself,  for his health and that of his family (in particular for children).

During and after the COVID 19 pandemic, this problem became a public debate and still remains a "difficult topic" to solve today.

Authors presents in details and with examples this problem and in conclusion they also express their opinion on this much debated topic.

This manuscript is adapted to special issue "ethical Dilemmas and moral distress in healthcare"

Pls only check english for little corrections.

Hence, this reviewer accept with minor revision this manuscript for publication in Healthcare.

Author Response

Please see tha attachment

Reviewer 4 Report

Line 59 : haemophiles influenzae -> haemophilus influenza (when you refer to a certain pathogen using the name of the gender and species in latin, it should be best written in italics)

Line 60 – 61 : Although, [However : fits better here] it should be noted that the Ministry of Health in Israel does not have central statistics, and usually presents data on vaccination rates in districts or regions.

Line 62 – 75 : All the statistics that are presented here refer to the vaccination coverage between the years 2017 and 2021 ? need to be a little more clear with each sentence.  

Line 75 : and in the case of extreme production of diseases that may return -> not clear what the authors mean here

Line 83-85 : In Israel, the question of general vaccinations of children against diseases like measles, mumps and chickenpox has become unquestionably and unpleasantly "popular" in the negative sense. -> the meaning of this period is not clear

Line 125 – 126 : were reasons that reflected difficulty -> not clear what the authors mean by difficulty…

Line 134 : "damage to the children's immune system -> the second “ is missing

Line 134 – 139 : Some parents believe that they can protect their children from exposure to a contagious disease through hygiene, good nutrition and a healthy lifestyle, and have a preference for natural healing methods, and believe that natural immunization (allowing a child to get the disease) is preferable to getting an artificial vaccine. It is obvious that even parents who do vaccinate their children regularly have doubts about the effectiveness of vaccines. -> are there statistics for those declarations as well

Legal Solutions to Anti-Vaccination

When referring to countries that impose children’s vaccination in order for them to be enrolled to kindergarten or school, it would be better if the specific vaccines were also mentioned.

Line 268 : It can be a medical reason -> you could analyse some reasons a little further

Line 287-288 : In order to achieve substantial justice, people who cannot be vaccinated must be afforded a suitable environment -> what do you actually mean by “suitable environment” ?

Line 327-331 : Vaccination hesitancy is often not about lack of information but rather a surplus of information from varied (and frequently contradictory) sources, and attempts to deal with conflicts that vary from vaccine to vaccine such as vaccine safety and efficacy vs. the dangers of the disease, personal interests vs. contribution to the community, and so on -> lack of information – surplus of information has not been analyzed in the text yet it appears in the conclusion…
